# OpenReview forum: "Unlocking Noise-Resistant Vision: Key Architectural Secrets for Robust Models Against Gaussian Noise"
_ICML.cc/2026/Conference — ICML 2026 regular_

### Official Review · Reviewer_FaSV · 2026-03-05

**Soundness:** 3
**Presentation:** 3
**Significance:** 3
**Originality:** 3
**Overall Recommendation:** 4
**Confidence:** 5

**Summary:**

In the vision models robustness measurement task, the paper presents a comprehensive empirical and theoretical analysis. How specific architectural design choices in vision models influence robustness against additive Gaussian noise is presented. The empirical insights are converted into simple, actionable design rules. The robustness enhancement findings include larger stem kernels, smaller input resolutions, average pooling, and supervised vision transformers. Theoretical justifications for these empirical findings are presented via signal processing and statistical analysis of pooling operators. The vulnerability of CLIP ViTs is highlighted via a pixel-space Lipschitz bound.

**Compliance With Llm Reviewing Policy:**

Affirmed.

**Key Questions For Authors:**

1. The paper selects Gaussian noise over other corruptions. More citations and corruption-type evaluations should be discussed.
2. Is the vulnerability related to the contrastive loss?
3. What is the boundary of a smaller resolution that improves robustness? Any mathematical guarantee?

**Limitations:**

yes

**Strengths And Weaknesses:**

Strengths

Soundness: A large-scale meta-analysis of existing models with controlled experiments to isolate variables is presented. This dual approach strengthens the validity of their claims. The theoretical results are mathematically consistent with standard signal processing theory.

Presentation: The paper is well-written and organized.

Significance: The robustness enhancement findings include larger stem kernels, smaller input resolutions, average pooling, and supervised vision transformers

Originality: The concept of deep learning robustness analysis is new.

Weaknesses

Soundness: The rigorous isolation of the normalization factor in CLIP models should be described in more detail.

Presentation: The overall evaluation pipeline is not shown in a figure form. Open-sourced code, not just some Python files, is not provided.

Significance: The significance is limited by the exclusive focus on Gaussian noise, which can not fully represent semantic corruptions or geometric transformations.

Originality: The mathematical derivations are standard results in statistics. The application for explaining the robustness gap between modern architectures should be further explained. The specific attribution of CLIP fragility to its normalization constants needs further emphasis.

---

> ### Author Rebuttal · Authors · 2026-03-30
>
> Thank you for your valuable feedback on our manuscript.
>
> W3, Q1 (Gaussian Noise) We clarify our claim as follows: Gaussian noise is a conservative probe for a broad family of corruptions under the assumptions outlined in Appendix G, rather than a literal model of every real-world perturbation. Appendix G gives the corresponding worst-case, upper-envelope result, and Figure 7 evaluates blur, weather, and digital corruptions. Appendix H further discusses extensions beyond Gaussianity, including Poisson and salt-and-pepper noise. We will revise the manuscript to state this rationale and scope more clearly.
>
> W4, Q2 (CLIP) We performed controlled ablation experiments to isolate possible causes of the CLIP gap, including factors that might arise from contrastive pretraining. Note that if properties specific to the CLIP objective or pretrained weights were the dominant cause, swapping weights with supervised ViTs would be expected to remove the vulnerability. However, our ablation studies show that weight swapping does not remove the gap (Tables 17, 18, and Appendix F), whereas changing the normalization constants yields much larger and more consistent improvements. We will revise the manuscript to state more explicitly that normalization is the dominant factor supported by our experiments, while avoiding making a stronger claim about the training objective itself.
>
> Q3 (Smaller Resolution) Of course, there is a boundary beyond which further downsampling to make a smaller resolution becomes detrimental, so our claim is valid only within that regime, rather than following a monotone "smaller is always better" rule. To answer the reviewer's boundary question, we also ran an additional sweep on ImageNet-1K validation images using `vit_base_patch16_224.augreg_in1k` and `vit_base_patch16_384.augreg_in1k`. We swept 160/192/224/256/288/320/352/384 at $\sigma$=0.02/0.05/0.10.
>
> | Checkpoint | Best consistency res. `(0.02/0.05/0.10)` | Best noisy top-1 res. `(0.02/0.05/0.10)` | Mean near-optimal band |
> | --- | --- | --- | --- |
> | `224` ckpt | `192/224/192` | `320/256/288` | `224/256/288/320` |
> | `384` ckpt | `256/320/320` | `352/384/352` | `352/384` |
>
> From this result, we conclude that the boundary is the lower edge of the near-optimal band, not a universal pixel value. For the 224 checkpoint, the smallest resolution within 0.5 pp of the mean optimum is 224; going lower hurts consistently (224-160 = +4.20/+5.08/+5.27 pp noisy top-1; 224-192 = +1.61/+1.66/+1.56 pp). For the 384 checkpoint, the lower boundary is 352 rather than a smaller resolution (352-320 = +0.98/+1.07/+0.78 pp; 352-224 = +5.32/+5.32/+5.08 pp). Thus, there is no checkpoint-independent rule such as "smaller is always better."
>
> We present a corresponding mathematical interpretation here. Theorem 4.3 gives $\gamma_{\downarrow}(s)\le C'/s^2$; we have $\mathbb{E}\|D_s\eta\|_2^2 \le \sigma^2 HWC'/s^4$. If the resized classifier is $L_s$-Lipschitz and has a clean logit margin $m_s(x)$, then a standard margin certificate plus Markov gives $\Pr[\text{flip at resolution }s] \le 4L_s^2\sigma^2 HWC'/(m_s(x)^2 s^4)$. This gives an explicit boundary condition for our pre-resize protocol: downsampling is beneficial only while the $s^{-4}$ noise reduction dominates the loss in $m_s(x)$. Because $m_s(x)$ is image- and checkpoint-dependent, there is no universal cutoff, which is exactly what the new sweep shows. We will revise the manuscript accordingly to reflect this point.
>
> W1, W2 (Presentation) Thank you for the kind suggestion, and we agree that the presentation can be improved. Following the reviewer's suggestion, we will add a figure to illustrate the full evaluation pipeline and release the source code used in this study.

---

> > ### Author Rebuttal · Reviewer_FaSV · 2026-04-01
> >
> > The source code problem is not solved.

---

> > > ### Author Response · Authors · 2026-04-04
> > >
> > > Thank you for hearing from you. Following your suggestion, we have prepared our source code and results here. https://anonymous.4open.science/r/gaussian_noise_robustness-D1EC/
> > >
> > > We will make sure to release them with the manuscript. Thank you again for your thoughtful comments.

---

### Official Review · Reviewer_4HaG · 2026-03-11

**Soundness:** 3
**Presentation:** 3
**Significance:** 3
**Originality:** 3
**Overall Recommendation:** 5
**Confidence:** 4

**Summary:**

This paper presents a thorough and systematic investigation into the robustness of vision models against Gaussian noise. Through extensive empirical evaluations of 1,174 pre-trained models, the authors identify four critical micro-architecture design patterns associated with higher noise robustness: larger stem kernels, smaller input resolutions, average pooling, and the use of supervised ViTs over CLIP ViTs. Remarkably, the paper does not stop at empirical observation; it further develops a rigorous theoretical analysis framework, providing clear causal explanations for these empirical findings and offering actionable design principles. This work provides significant theoretical support and practical guidance for understanding and building more robust vision models.

**Compliance With Llm Reviewing Policy:**

Affirmed.

**Final Justification:**

The rebuttal addressed my concerns partially but constructively, and it clarified that these assumptions mainly support auxiliary interpretations rather than all core module-level results.
However, the rebuttal did not fully resolve my central concern, because it still did not provide a more explicit discussion of the validity range of these assumptions.
Overall, the rebuttal made the paper’s claims clearer but did not change my score, and I still believe the paper merits acceptance.

**Key Questions For Authors:**

See weaknesses above.

**Limitations:**

See weaknesses above.

**Strengths And Weaknesses:**

Strengths:
1.Robustness is a core challenge in deep learning. While existing research often focuses on improving robustness, this paper aims to understand the intrinsic mechanisms behind the differing robustness of existing architectures. By delving into micro-architecture design choices, the paper reveals the "secrets" hidden within these architectures. This exploration of fundamental mechanisms holds deeper academic value than merely pursuing performance gains.
2.The paper systematically evaluates 1,174 pre-trained models from the timm library. The extensive experimental work and robust evidence base ensure the generality and statistical significance of the conclusions.
3.The paper distills complex theoretical analyses into "simple and actionable design rules." These rules are intuitive and easy to apply, offering inspiring directions for researchers in vision architecture design.
4.The paper's most significant contribution lies in elevating empirical observations to a theoretical level, revealing the physical and mathematical principles behind these design choices. This establishes not just a correlation but a causal mechanism between design patterns and robustness.
5.The paper presents clear theorem statements and proof sketches, with complete derivations provided in the appendix, demonstrating a solid theoretical foundation.

Weaknesses:
1.Although the paper argues for the validity of using Gaussian noise as a proxy, real-world image perturbations are far more complex than Gaussian noise alone. Perturbations such as structured noise, adversarial perturbations, specific sensor artifacts, or combinations of multiple degradations may exhibit significantly different statistical properties. The paper mentions this in its introduction and limitations section. It is suggested that future work could further investigate the effectiveness and applicability boundaries of the proposed micro-architecture design principles under a wider and more complex range of perturbation types.
2.Parts of the theoretical analysis rely on assumptions like "small perturbations" and "local linearization." While these are common analytical tools, the precise scope of these assumptions and their impact on the conclusions under the strong non-linearity of deep neural networks warrant a more in-depth discussion. For instance, when the input noise intensity is high, higher-order terms may no longer be negligible. How precise are the conclusions in such scenarios? A more detailed analysis of this issue would further strengthen the universality of the theory.

---

> ### Author Rebuttal · Authors · 2026-03-30
>
> Thank you for your kind words and valuable suggestions. We address the concerns of the reviewer below.
>
> W1 (Other Perturbations) We agree that this point needs clarification. Our claim is not that Gaussian noise literally models every real-world perturbation, but rather that it serves as a conservative probe or worst-case bound for a family of corruptions under the assumptions stated in Appendix G. Furthermore, Figure 7 in Appendix G also reports results on blur, weather, and digital corruptions, which support this interpretation. Appendix H further shows that our findings are not tied to Gaussianity itself: we discuss how analogous conclusions extend to other noise models, such as Poisson and salt-and-pepper noise. We will revise the manuscript to state this scope more clearly and to point readers explicitly to Appendices G and H.
>
> W2 (Noise Scale and Local Linearization) We agree that the scope of the small-perturbation and local-linearization assumptions should be stated more clearly. These assumptions are needed for the auxiliary analyses that interpret Gaussian noise as a conservative probe and RankDiff as a local robustness slope, but we ensure you that our core module-level results in Theorems 4.1, 4.3, 5.1, and 6.1 are free from these assumptions. We will revise the paper to more clearly distinguish these statements from the parts that rely on local linearization. Also, when local linearization was applied, we ensured to state the higher-order remainder terms explicitly via big-O bounds; we will make the corresponding validity range and interpretation more explicit in the revision.

---

> > ### Author Rebuttal · Reviewer_4HaG · 2026-04-06
> >
> > Thank you for the rebuttal. The response addresses my concerns in a constructive way, but they are only partially resolved. For the theoretical part, I appreciate the clarification that the small-perturbation and local-linearization assumptions mainly support auxiliary interpretations rather than all core module-level results. However, the rebuttal does not yet provide a more explicit quantitative discussion of the validity range of these assumptions, especially under stronger noise where higher-order terms may become non-negligible. Overall, while the rebuttal is helpful and reduces some of my concerns, I do not think it fully resolves the central issues raised in my previous review.

---

### Official Review · Reviewer_8vur · 2026-03-12

**Soundness:** 2
**Presentation:** 3
**Significance:** 2
**Originality:** 2
**Overall Recommendation:** 3
**Confidence:** 3

**Summary:**

The paper studies robustness of vision models to additive Gaussian noise by evaluating 1,174 pretrained models from timm on ImageNet-1K under clean and noisy inputs, and summarizes robustness via a rank-based metric (“RankDiff”). Beyond the large-scale empirical study, the paper provides theoretical analyses aimed at explaining systematic robustness trends associated with architectural and design choices. In particular, it studies the effect of larger stem kernels / larger ViT patch sizes, smaller input resolutions, average pooling (vs. max pooling), and supervised ViTs (vs. CLIP-pretrained ViTs). The authors attempt to explain these trends theoretically, linking anti-aliasing/downsampling and input standardization scaling factors to how Gaussian noise reduction.

**Compliance With Llm Reviewing Policy:**

Affirmed.

**Final Justification:**

I thank the authors for the detailed rebuttal and for engaging seriously with my comments. While these clarifications improve the precision of the paper, I am still not fully convinced that the resulting findings are sufficiently significant to justify a higher overall score.

**Key Questions For Authors:**

1. At what exact stage in the preprocessing pipeline is Gaussian noise injected for models with different native input resolutions? In case it was added before preprocessing, how would this affect the reported resolution-robustness trend if noise were instead injected after resizing (i.e., at the model’s native input size)?

2. Can the authors separate the effect of the preprocessing resize from intrinsic architectural differences? For example, can you evaluate all models under a unified resizing / pre-filtering scheme (e.g., standardize to 224 first, then upsample to 384 where applicable), and report how much the RankDiff conclusions change?

**Limitations:**

Yes

**Strengths And Weaknesses:**

**Strengths**

1. The paper conducts an extensive empirical study covering more than 1,000 large-scale pretrained models. This breadth provides a global picture of robustness across architectural families, model sizes and training techniques.

2. The paper provides theoretical justification for focusing on Gaussian noise, arguing (under a local linearization / variance-bounded corruption model) that Gaussian perturbations serve as a convenient worst-case probe for a broad family of common corruptions. The signal-processing perspective used to analyze architectural components is technically sound under the stated assumptions.

3. The paper provides concrete and practically useful recommendations. For example, it shows that simple preprocessing adjustments when fine-tuning CLIP-based models can significantly improve robustness, without modifying the architecture itself.


**Weaknesses**

1. In several cases, the reported robustness gaps appear to stem primarily from preprocessing and input scaling choices rather than architectural or training differences.

    1. The paper reports that smaller input resolutions correlate with better Gaussian noise robustness. However, it is not fully clear where noise is injected relative to the input resizing pipeline across models with different native resolutions. If noise is added before resizing, downsampling/interpolation may implicitly attenuate noise energy, which could explain much of the improved robustness of low-resolution models. This interpretation is consistent with the paper’s own analysis in Section 4.2, where anti-aliased downsampling is shown to reduce noise energy.
    While injecting noise before preprocessing may seem like a fair evaluation protocol, attributing the resulting differences to the models themselves may be misleading in this case, since simple preprocessing remedies (e.g., explicit pre-filtering, alternative resizing strategies, or injecting noise after resizing) could possibly close the gap without changing the model.
    2. The paper attributes the vulnerability of CLIP-based models primarily to their normalization scaling (Section 6), showing that a simple change in preprocessing substantially alters robustness. While this is an interesting and useful finding, it weakens the broader conclusion that CLIP training objectives inherently produce less robust representations; the evidence suggests a large fraction of the effect is explained by pixel scaling conventions rather than representational differences.

2. Although the paper’s mathematical formulation connecting anti-aliasing to noise reduction is somewhat novel in this context, the broader claim that aliasing / downsampling choices materially affect stability and robustness has been widely discussed in the ML vision literature (e.g., [1–3]).

 [1] Impact of Aliasing on Generalization in Deep Convolutional Networks, Vasconcelos et al., 2021

 [2] How Convolutional Neural Networks Deal with Aliasing, Ribeiro & Schön, 2021

 [3] Aliasing and Adversarial Robust Generalization of CNNs, Grabinski et al., 2022

---

> ### Author Rebuttal · Authors · 2026-03-30
>
> We thank the reviewer for sharing insightful comments.
>
> Q1, Q2. (Noise-injection stage and intrinsic backbone effect) In our protocol, Gaussian noise is added to the raw image before any preprocessing:
>
> Raw image -> Add Gaussian noise -> Resize -> CenterCrop -> ToTensor -> Mean-std normalization -> Model input
>
> We chose this protocol because it naturally describes sensor noise in real imaging pipelines, which is thereby applied to the raw image before resizing and normalization; subsequent preprocessing is deterministic. We nevertheless agree that, under this sensor-noise protocol, the advantage of lower native resolution can mix two effects: downsampling and intrinsic backbone properties.
>
> To decouple them, we ran matched-pair counterfactuals on 7 supervised 224-vs-384 pairs across Flowers102, Oxford-IIIT Pet, and ImageNet-1K, using three conditions: `pre_native` (noise added before each native resize), `post_native` (noise added after each native resize), and `post_u224` (shared 256->224 pipeline, then 224->384 upsampling for the 384 model, with noise added after the final resize). We measured top-1 consistency; for ImageNet, we also measured noisy top-1 accuracy. Relative logit change and JS divergence showed the same pattern.
>
> For the original matched pair `vit_base_patch16_224.augreg_in1k` versus `vit_base_patch16_384.augreg_in1k`, the consistency gap on ImageNet changes from `-0.29/+0.34/+0.24` in `pre_native` to `+1.56/+2.73/+3.91` in `post_native`, and the noisy top-1 gap becomes `+1.07/+1.76/+3.81` points in favor of 384. Thus, once the preprocessing confound is removed, the direction of the effect changes.
>
> | Mean consistency gap (384-224, pts), # pos./9 | pre_native | post_native | post_u224 |
> | --- | ---: | ---: | ---: |
> | ViT-B/16 augreg_in1k | -0.67 (5/9) | +5.47 (9/9) | +3.56 (8/9) |
> | ViT-B/16 augreg_in21k_ft_in1k | +1.27 (9/9) | +3.73 (9/9) | +0.73 (7/9) |
> | ViT-B/32 augreg_in1k | -2.22 (3/9) | +3.69 (8/9) | -1.17 (5/9) |
> | ViT-B/32 augreg_in21k_ft_in1k | +0.73 (6/9) | +6.25 (9/9) | +3.40 (8/9) |
> | ViT-B/16 orig_in21k_ft_in1k | -1.10 (5/9) | +3.40 (9/9) | +1.52 (8/9) |
> | DeiT3-B/16 fb_in1k | -1.60 (2/9) | +1.96 (9/9) | +1.76 (8/9) |
> | DeiT3-B/16 fb_in22k_ft_in1k | +0.04 (4/9) | +4.74 (9/9) | -0.64 (3/9) |
> | Mean over 7 pairs | -0.50 (34/63) | +4.18 (62/63) | +1.31 (47/63) |
>
> | IN1K noisy acc gain (384-224, pts), # pos./3 | post_native | post_u224 |
> | --- | ---: | ---: |
> | ViT-B/16 augreg_in1k | +2.21 (3/3) | +3.16 (3/3) |
> | ViT-B/16 augreg_in21k_ft_in1k | +1.53 (3/3) | +1.94 (3/3) |
> | ViT-B/32 augreg_in1k | +4.33 (3/3) | +4.35 (3/3) |
> | ViT-B/32 augreg_in21k_ft_in1k | +3.92 (3/3) | +3.14 (3/3) |
> | ViT-B/16 orig_in21k_ft_in1k | +2.95 (3/3) | +1.74 (3/3) |
> | DeiT3-B/16 fb_in1k | +1.95 (3/3) | +0.34 (2/3) |
> | DeiT3-B/16 fb_in22k_ft_in1k | +2.26 (3/3) | +1.63 (3/3) |
> | Mean over 7 pairs | +2.74 (21/21) | +2.33 (20/21) |
>
> These controls support the reviewer's concerns and sharpen the claim. Under our pre-resize sensor-noise protocol, a substantial part of the low-resolution advantage should be interpreted as a downsampling effect. Although the current manuscript still attributes the advantage of smaller resolution to this downsampling effect (Theorems 4.1 and 4.3), we will revise the manuscript to clearly state that this advantage is not due to intrinsic architectural robustness. Across the 7-pair extension, `post_native` favors 384 in 62/63 consistency cells and 21/21 ImageNet noisy-accuracy cells, and `post_u224` favors 384 in 47/63 consistency cells and 20/21 noisy-accuracy cells. We also checked same-checkpoint controls under `post_u224`: simply instantiating the same 224 checkpoint at 384 does not create a robustness gain, further arguing against an intrinsic advantage of lower input size itself.
>
> W1-2 (CLIP) We agree that our original wording should be more precise. Appendix F shows that normalization constants explain a large fraction of the observed CLIP fragility: for example, on the Oxford-IIIT Pet, `vit_base_patch16_clip_224.openai` improves noisy test accuracy from 70.2 to 77.3 when only the preprocessing constants are changed from `OPENAI` to `INCEPTION`. By contrast, changing CLIP-specific properties, such as weight swapping and `norm_pre` controls do not remove the gap, as demonstrated in the ablation of Tables 17, 18, and 19 in Appendix F. We will therefore avoid implying that the CLIP objective itself is the primary cause and will instead clearly state that normalization constants explain a major portion of the effect.
>
> W2 (Related Work) Thank you for the suggestion. We will better position the paper relative to prior aliasing literature [1-3]. Indeed, our intended novelty is not the generic claim that aliasing matters, but rather the explicit noise-attenuation predictions for downsampling choices together with large-scale confirmation across 1,174 timm models.

---

> > ### Author Rebuttal · Reviewer_8vur · 2026-04-02
> >
> > I thank the authors for the detailed rebuttal and for conducting additional experiments that directly address my questions. I appreciate both the effort and the clarity of the response. At the same time, the new results reinforce my main concern regarding two of the paper’s main conclusions. In particular, the additional results suggest that, under the evaluation protocol used in the manuscript, the apparent robustness advantage of lower input resolution largely reflects the preprocessing/downsampling pipeline rather than an intrinsic robustness property of the backbone. Likewise, the rebuttal clarifies that a major portion of the observed CLIP vulnerability is explained by normalization constants, which weakens a broader interpretation in terms of the training objective or representations themselves. These findings remain useful, but in my view they reduce the significance of the manuscript’s main claims. Therefore, I do not plan to change my original score.

---

> > > ### Author Response · Authors · 2026-04-04
> > >
> > > Thank you for the clarification and for acknowledging that our rebuttal addressed the technical concerns. We agree that the additional experiments clarify and sharpen the interpretation of two claims, and in the revised manuscript we will state this explicitly. In particular, under the pre-resize sensor-noise protocol, the apparent advantage of lower native resolution should be interpreted primarily as a downsampling effect rather than an intrinsic backbone property; likewise, a major portion of the observed CLIP fragility is explained by normalization constants rather than being attributed broadly to the CLIP objective itself.
> > >
> > > That said, we respectfully believe these clarifications sharpen the paper’s contribution rather than diminish it. The main contribution is rather a large-scale and modular analysis of Gaussian noise robustness across 1,174 timm models, together with theory and controlled experiments that identify which components of the end-to-end pipeline materially affect robustness. In this revised framing, the paper provides: (1) a realistic sensor-noise evaluation perspective, (2) matched-pair evidence that separates downsampling effects from backbone effects, (3) a clear empirical and theoretical account of pooling and normalization choices, and (4) practical guidance on how robustness can be improved either architecturally or through preprocessing.
> > >
> > > We will make sure that the final version avoids overstating intrinsic low-resolution robustness or attributing the CLIP gap primarily to the training objective. We hope the paper will be evaluated based on this clarified and more precise contribution.

---

### Official Review · Reviewer_pE8L · 2026-03-13

**Soundness:** 2
**Presentation:** 3
**Significance:** 3
**Originality:** 3
**Overall Recommendation:** 4
**Confidence:** 3

**Summary:**

This paper investigates how to improve model robustness to Gaussian noise from the perspective of key architectural design components in neural networks. Through extensive empirical experiments combined with a certain level of theoretical analysis, the authors summarize and validate several empirical observations. The theoretical and experimental analyses may provide useful guidance for the design and selection of model architectures in current practice.

**Compliance With Llm Reviewing Policy:**

Affirmed.

**Final Justification:**

The rebuttal solved most of my concerns, thus, I am raising my score.

**Key Questions For Authors:**

Please see the weakness

**Limitations:**

yes

**Strengths And Weaknesses:**

# strengths
1. Extensive experiments. The paper conducts a large number of experiments to validate various design choices, covering a broad range of settings.
2. Clear writing and structure. The paper is well organized and easy to follow.
3. Interesting findings. Some of the conclusions are insightful and may offer practical guidance for current industry practices.

# weakness
1. Limited generalizability of conclusions. The theoretical analysis relies on several assumptions, which raises questions about the generality of the conclusions. In addition, the evaluation metrics are somewhat limited and rely heavily on specific metrics and tasks, which may not comprehensively measure model performance. For example, the kernel analysis depends on the low-pass envelope assumption, which is not strictly justified.
2. The claim that lower resolution leads to better robustness may lack universality. This observation may be influenced by dataset characteristics and could depend on factors such as information loss due to downsampling, the spectral properties of the dataset, and training configurations. Further validation across more diverse domains would be necessary.
3. Conditional theoretical analysis. The theoretical analysis itself is also conditional, and the overall conclusions are largely driven by empirical observations rather than rigorous theoretical guarantees.

---

> ### Author Rebuttal · Authors · 2026-03-30
>
> Thank you for your valuable comments.
>
> W1, W3. The radial low-pass envelope assumption is used only for the kernel- and resolution-scaling laws discussed in Section 4.2, not for the others such as the Gaussian-envelope theorem, pooling analysis, or the normalization result. Appendix L and Table 20 also validate the assumption empirically: for representative box and Gaussian kernels, the measured radial spectra stay below the fitted envelope across multiple radii. Thus, only part of our theory depends on this assumption, and for that part, we provide an explicit empirical check.
>
> W1, W2. We agree that the original wording regarding resolution should be made more precise. In our protocol, Gaussian noise is injected at the raw-image stage before preprocessing:
>
> Raw image -> Add Gaussian noise -> Resize -> CenterCrop -> ToTensor -> Mean-std normalization -> Model input
>
> This protocol naturally describes sensor noise in real imaging; subsequent preprocessing is deterministic. Nevertheless, as the reviewer correctly pointed out, we should decouple the effects of the input pipeline from the intrinsic backbone properties. Following the reviewer's suggestion, we performed additional experiments to test this directly.
>
> Specifically, we performed counterfactual experiments on the matched pair `vit_base_patch16_224.augreg_in1k` and `vit_base_patch16_384.augreg_in1k` over Flowers102, Oxford-IIIT Pet, and ImageNet-1K. We compared the original setting `pre_native` (noise added before each model's native resize) with two post-resize controls: `post_native` (noise added after each native resize) and `post_u224` (shared 256->224 pipeline, then 224->384 upsampling for the 384 model, with noise injected after the final resize). We measured top-1 consistency, and for ImageNet, we also measured noisy top-1 accuracy.
>
> | Top-1 consistency gap (384-224, pts) | $\sigma$=.02 | .05 | .10 |
> | --- | ---: | ---: | ---: |
> | Flowers / pre_native | -6.05 | -2.83 | +1.76 |
> | Flowers / post_native | +8.59 | +11.72 | +11.23 |
> | Flowers / post_u224 | +6.10 | +8.20 | +10.89 |
> | Pet / pre_native | -0.73 | +0.54 | +1.03 |
> | Pet / post_native | +2.00 | +2.78 | +4.69 |
> | Pet / post_u224 | +0.49 | -1.12 | +1.95 |
> | IN1K / pre_native | -0.29 | +0.34 | +0.24 |
> | IN1K / post_native | +1.56 | +2.73 | +3.91 |
> | IN1K / post_u224 | +0.39 | +0.83 | +4.30 |
>
> | IN1K noisy top-1 acc gain (384-224, pts) | $\sigma$=.02 | .05 | .10 |
> | --- | ---: | ---: | ---: |
> | post_native | +1.07 | +1.76 | +3.81 |
> | post_u224 | +2.15 | +2.39 | +4.93 |
>
> A same-checkpoint `post_u224` control also does not create a robustness gain simply by instantiating the same checkpoint at 384, which is again inconsistent with an intrinsic low-resolution advantage.
>
> | Mean consistency gap (384-224, pts), # pos./9 | pre_native | post_native | post_u224 |
> | --- | ---: | ---: | ---: |
> | ViT-B/16 augreg_in1k | -0.67 (5/9) | +5.47 (9/9) | +3.56 (8/9) |
> | ViT-B/16 augreg_in21k_ft_in1k | +1.27 (9/9) | +3.73 (9/9) | +0.73 (7/9) |
> | ViT-B/32 augreg_in1k | -2.22 (3/9) | +3.69 (8/9) | -1.17 (5/9) |
> | ViT-B/32 augreg_in21k_ft_in1k | +0.73 (6/9) | +6.25 (9/9) | +3.40 (8/9) |
> | ViT-B/16 orig_in21k_ft_in1k | -1.10 (5/9) | +3.40 (9/9) | +1.52 (8/9) |
> | DeiT3-B/16 fb_in1k | -1.60 (2/9) | +1.96 (9/9) | +1.76 (8/9) |
> | DeiT3-B/16 fb_in22k_ft_in1k | +0.04 (4/9) | +4.74 (9/9) | -0.64 (3/9) |
> | Mean over 7 pairs | -0.50 (34/63) | +4.18 (62/63) | +1.31 (47/63) |
>
> | IN1K noisy acc gain (384-224, pts), # pos./3 | post_native | post_u224 |
> | --- | ---: | ---: |
> | ViT-B/16 augreg_in1k | +2.21 (3/3) | +3.16 (3/3) |
> | ViT-B/16 augreg_in21k_ft_in1k | +1.53 (3/3) | +1.94 (3/3) |
> | ViT-B/32 augreg_in1k | +4.33 (3/3) | +4.35 (3/3) |
> | ViT-B/32 augreg_in21k_ft_in1k | +3.92 (3/3) | +3.14 (3/3) |
> | ViT-B/16 orig_in21k_ft_in1k | +2.95 (3/3) | +1.74 (3/3) |
> | DeiT3-B/16 fb_in1k | +1.95 (3/3) | +0.34 (2/3) |
> | DeiT3-B/16 fb_in22k_ft_in1k | +2.26 (3/3) | +1.63 (3/3) |
> | Mean over 7 pairs | +2.74 (21/21) | +2.33 (20/21) |
>
> In short, the reviewer's concern is correct: Under our pre-resize sensor-noise protocol, the observed advantage of native resolution should be interpreted as a downsampling effect. Although the current manuscript still attributes the advantage of smaller resolution to this downsampling effect (Theorems 4.1 and 4.3), we will revise the manuscript to clearly state that this advantage is not due to intrinsic architectural robustness. Indeed, once the noise is moved after resizing, we find no evidence that the 224 model is intrinsically more robust than the 384 model; across the 7-pair extension, the `post_native` favors the 384 in 62/63 consistency cells and 21/21 ImageNet noisy-accuracy cells, and `post_u224` favors the 384 in 47/63 consistency cells and 20/21 noisy-accuracy cells. We will therefore revise the abstract, introduction, Section 4.2 discussion, and conclusion by replacing the broad claim that "smaller input resolution is more robust" with the correct, precise statement supported by the new results.

---

### Decision · Program_Chairs · 2026-04-30

**Decision:**

Accept (regular)

**Comment:**

The paper studies the impact of architectural design choices on model robustness to Gaussian noise. The reviews agree that the empirical study is interesting and provides some new insights. The rebuttal provides extensive additional experiments and addresses concerns regarding presentation and significance. However, there are remaining concerns regarding the disambiguation of effects of the the architectural design and effects of the pre-processing pipeline (for example downsampling). Weighing points raised in the reviews and the rebuttal, I still think the paper makes an interesting contribution. The paper would strongly benefit from a detailed discussion of potential confounding factors from data pre-processing.